# Effect of Physical and Enzymatic Modifications on Composition, Properties and In Vitro Starch Digestibility of Sacred Lotus (*Nelumbo nucifera*) Seed Flour

**DOI:** 10.3390/foods11162473

**Published:** 2022-08-17

**Authors:** Pornnutcha Sopawong, Daruneewan Warodomwichit, Warangkana Srichamnong, Pawadee Methacanon, Nattapol Tangsuphoom

**Affiliations:** 1Joint Graduate Program in Nutrition, Faculty of Medicine Ramathibodi Hospital, Mahidol University, Bangkok 10440, Thailand; 2Division of Nutrition and Biochemical Medicine, Department of Medicine, Faculty of Medicine Ramathibodi Hospital, Mahidol University, Bangkok 10440, Thailand; 3Food and Nutrition Academic and Research Cluster, Institute of Nutrition, Mahidol University, Nakhon Pathom 73170, Thailand; 4Food Materials Research Team (FOMT), Advanced Polymer Technology Research Group, National Metal and Materials Technology Center (MTEC), Pathum Thani 12120, Thailand

**Keywords:** sacred lotus, lotus seed flour, physical modification, enzymatic modification, starch digestibility

## Abstract

In this study, native lotus seed flour (N-LSF) was modified by different methods, namely, partial gelatinization (PG), heat–moisture treatment (HMT), or pullulanase treatment (EP). Their composition, functional properties, starch composition, and estimated glycemic index (eGI) were compared. PG contained similar protein, soluble dietary fiber, and insoluble dietary fiber contents to N-LSF, while those of HMT and EP differed from their native form. PG increased rapid digestible starch (RDS) but decreased resistant starch (RS); while HMT and EP increased amylose and RS contents to 34.57–39.23% and 86.99–92.52% total starch, respectively. Such differences led to the different pasting properties of the modified flours rather than PG, which was comparable to the native flour. HMT had limited pasting properties, while EP gave the highest viscosities upon pasting. The eGI of all samples could be classified as low (<50), except that of PG, which was in the medium range (60). It was plausible that lotus seed flour modified either with HMT or EP could be used as carbohydrate source for diabetes patients or health-conscious people.

## 1. Introduction

Diabetes mellitus (DM) is one of the metabolic diseases characterized by high fasting blood glucose levels, of which the prevalence and incidence are currently increasing glob-ally. It has been reported in 2021 that 537 million adults around the world are living with DM, and the number estimated in 2030 will rise to more than 643 million [1]. The percentage of deaths due to DM, as well as its complications, tend to increase and have a significant impact on hospital treatment expenditure for individuals, families, and societies. It is well-known that, in the long term, high blood glucose levels result in various complications, especially hypertension, diabetic ketoacidosis, stroke, and chronic damage of the eye, nerve, kidney, heart, and blood vessels. Therefore, low glycemic index (GI) food, which is slowly digested, absorbed, and metabolized, resulting in only slight fluctuations in blood glucose levels, is one of the criteria when selecting a carbohydrate food source for diabetic patients and health concerned people.

Lotus seed, particularly of sacred lotus (*Nelumbo nucifera*), is a rich source of starch, which provides carbohydrates as a nutrient and energy source for the human body. Many studies have reported that total starch content in lotus seed can reach over 60% on a dry basis [2,3,4]. Moreover, lotus seed has a low GI (<55), lower than that of waxy black rice (100), millet (93), Job’s tears (91), white rice (83) and dried lily bulb (83), but higher than adzuki bean (21) [5]. Moreover, it contains resistant starch (RS) (~40%), i.e., non-digestible but fermentable by probiotic bacteria in the colon, leading to the production of short-chain fatty acids such as acetic, butyric, and propionic acid. These short-chain fatty acids play an essential role in maintaining the balance of the intestinal microbiome that has a positive effect on health and prevents many diseases [6]. Zhang et al. [7] reported that RS from lotus seed could promote probiotic *Bifidobacterium adolescentis* growth and produce short chain fatty acids when tested in the simulated conditions of the human digestive system. Thus, lotus seed can be used as a prebiotic. In addition, lotus seed starch possessed low swelling power, and consequently high pasting and gelatinization temperatures, though the gelatinized granules are highly susceptible to enzymatic digestion. It also showed higher gel syneresis and higher ratio setback to peak viscosity during pasting, suggesting a stronger tendency for short-term retrogradation [2] when compared to starch of other plants, such as wheat [8], corn [9], cassava [10], rice [11], kidney bean [12], and yam [13,14]. With such specific characteristics, the lotus seed has gained an attention in food industry to formulate new products, especially as an alternative nutrient-rich and health-beneficial flour. Lotus seed flour has been used for substitution of wheat flour in cookies [15], bread [3], and noodles [16].

In general, native starches are inherently unsuitable for food applications. Therefore, they must be modified physically and/or chemically to enhance their positive attributes and/or to minimize their defects. Physical and enzymatic modifications are widely used in starch modification because the methods are considered as safe for human consumption, easy to apply, and considerably affects changes in starch or flour’s properties. Physical modification is mainly used to adjust the granular size, granular structure, crystallite region, the interaction between polysaccharide chains, or complex formation [17,18]. Many previous studies have demonstrated the physicochemical properties and expanded the application of modifications that improve and expand lotus seed starch applications, such as dry heat treatment [19], microwave heat treatment [20], retrograded [21], heat moisture treatment [22], and the combined methods [23,24,25,26].

However, to our knowledge, there is no report available on such properties of native and modified lotus seed flour, particularly with regard to their digestibility. In this study, therefore, native lotus seed flour was modified through physical and enzymatic methods, and their composition, physicochemical properties, and digestibility, as well as estimated glycemic index, were compared with those of native lotus seed flour. We expected that a thorough understanding of the influence of the modification processes on such properties of lotus seed flour would be beneficial in achieving our goal of broadening its application in food as an alternative flour with potential health benefits.

## 2. Materials and Methods

### 2.1. Raw Material and Preparation of Native Lotus Seed Flour

Sun-dried seeds of sacred lotus (*Nelumbo nucifera*) were purchased in four batches from a local farm located in Nakhon Sawan province, Thailand during July 2020 to September 2020. Once received, the seeds were sampled for moisture analysis, vacuum-packed in polyethylene plastic bags and stored at room temperature until being used for flour preparation. Moisture content of the lotus seeds ranged from 6.5–7% *w*/*w*. Prior to flour preparation, lotus seeds were soaked in deionized (DI) water (1:4 *w*/*v*, seed-to-water ratio) overnight, peeled, removed the embryos by hands, and dried in a hot air oven at 40 °C for 16–18 h to obtain dried lotus seeds with the final moisture content of 7–8% *w*/*w*. Native lotus seed flour (N-LSF) was prepared by grinding the dried seeds using a grinder mill machine, followed by sieving through a 100-mesh sieve. The flour was packed in polyethylene plastic bags and kept at room temperature until being used. The production yield of N-LSF from dried lotus seeds was 80–82%.

### 2.2. Physical Modifications of Lotus Seed Flour

#### 2.2.1. Partial Gelatinization

Dried lotus seeds were soaked in DI water (1:4 *w*/*v*, seed-to-water ratio) and kept under refrigeration at 4 °C for 12 h. They were parboiled at 75 °C for 10 min to achieve partial gelatinization of starch. Then, the parboiled seeds were dried at 60 °C in a hot air oven until the moisture content was 10–12% *w*/*w*. After drying, the seeds were ground and sieved through 100-mesh sieve to obtain partially gelatinized lotus seed flour (PG). The samples were packed in polyethylene plastic bags and kept in a desiccator. The yield of production of PG from dried lotus seeds was up to 80%.

#### 2.2.2. Heat–Moisture Treatment

Heat–moisture treated lotus seed flour (HMT) was prepared by slightly modifying the method of Whantongkhum and Suwannaporn [27]. DI water was added to N-LSF at the flour-to-water ratio of 1:4 *w*/*v*. The mixture was kept overnight at 4 °C. After that, the excess water was drained out of the equilibrated slurry by vacuum suction. The slurry was then dried at 40 °C in a hot air oven for 12–14 h until its moisture content dropped to the desired level of 25% *w*/*w*, before being heated at 100 °C in a hot air oven for 2 h to obtain the final moisture content of 10–12% *w*/*w*. The obtained HMT samples were packed in polyethylene plastic bags and stored in a desiccator until required for further analysis. The recovery of HMT from N-LSF was about 75%.

### 2.3. Enzymatic Modification of Lotus Seed Flour

Enzymatic modification of lotus seed flour was performed using pullulanase according to the method of Wattananapakasem et al. [28] with slight modifications. A slurry (10% *w*/*v*) of N-LSF in DI water was prepared. The pH of the slurry was adjusted to 4.5 with 0.1 M sodium acetate buffer solution pH 4.5. Pullulanase (1000 NPUN/g, E2412, Sigma-Aldrich, St. Louis, MO, USA) was added to the slurry at an enzyme-to-substrate ratio of 20 NPUN/g dry flour. Enzymatic treatment was performed at 55 °C for 6 h in a shaking water bath. The treated slurry was then heated at 70 °C for 10 min to stop the enzyme reaction. Sample was centrifuged at 4600× *g* for 10 min, and the precipitate was washed twice with DI water. The precipitate was collected and dried at 45 °C in a hot air oven until the final moisture content reached 10–12% *w*/*w* to obtain the enzymatically modified flour (EP). The recovery of EP from N-LSF was about 70%.

### 2.4. Characterization of Lotus Seed Flour

#### 2.4.1. Chemical Composition

Chemical composition of native and modified lotus seed flours was determined according to the AOAC Official Methods [29]. Moisture content was measured using the gravimetric method (AOAC Method 925.10), protein content was analyzed as nitrogen content using the Kjeldahl method and a conversion factor of 5.70 (AOAC Method 991.20). Fat content was analyzed by acid hydrolysis (AOAC Method 922.06). Ash content was determined by incineration (AOAC Method 930.30). Total carbohydrate content was calculated by subtracting the contents of moisture, protein, fat, and ash from 100. Soluble and insoluble dietary fiber was determined using enzyme-gravimetric method (AOAC Method 941.43).

#### 2.4.2. Amylose Content

Amylose content was determined using the iodine absorption method [30] with slight modifications. Lotus seed flour (10 mg) was dissolved in 99.8% Ethanol (90 μL) and 1 M aqueous NaOH (540 μL). The mixture was vortexed and left at room temperature overnight. An aliquot of the sample was diluted with DI water to the final concentration of 1 mg starch/200 μL concentration. The diluted sample (200 μL) was neutralized with 0.05 M citric acid (1 mL) before addition of iodine solution (0.2 g I_2_ + 2 g KI in DI water, 800 μL). A final volume was then made up to 12 mL with DI water. The sample was immediately mixed and chilled in the refrigerator for 20 min before test. Amylose content was determined using 96-well UV visible microplate reader spectrophotometer (Synergy HT, Biotek Instrument, Winooski, VT, USA) at the wavelength of 620 nm. The absorbance values were converted to percent of amylose by standard curve of amylose from potato (2–12 mg, Merck KGaA, Darmstadt, Germany).

#### 2.4.3. Starch Fractions

Rapid digestible starch (RDS) and slow digestible starch (SDS) contents were analyzed based on in vitro starch digestibility procedure [31]. Digesta was taken at 20 and 120 min of α-amylase incubation. The glucose content was measured using an assay kit GOPOD-format K-GLUC 08/18 (Megazyme, Bray, Ireland). RDS and SDS were calculated from glucose content released from the sample after 20 and 120 min of digestion, according to the following equations:RDS = G20 × 0.9(1)
SDS = (G120 − G20) × 0.9(2)
where G20 is glucose content after 20 min of digestion, G120 is glucose content after 120 min of digestion, and 0.9 is glucose to starch conversion factor.

Resistant starch (RS) content was analyzed based on an in vitro starch digestibility procedure [32] with slight modifications. After 2 h of α-amylase incubation, the sample was centrifuged at 4600× *g* for 10 min. The residue as washed with 10 mL of DI water and recentrifuged at the same condition. Then, 3 mL of DI water and 3 mL of 4 M KOH was added and the mixture was incubated at room temperature for 30 min. After that, 5.5 mL of 2 M HCl, 3 mL of 0.4 M sodium acetate buffer pH 4.75 and 80 μL (3.26 U) of amyloglucosidase in 0.4 M sodium acetate buffer pH 4.75 were added, followed by vortexing. The mixture was incubated at 60 °C for 45 min in a shaking water bath. Next, it was centrifuged at 4600× *g* for 10 min to collect the supernatant. The residue was washed with 10 mL of DI water and centrifuged again. The supernatant from the first and second extractions were combined for the measurement of glucose content using glucose oxidase–peroxidase kit GOPOD reagent enzymes (K-GLUC 08/18, Megazyme).

#### 2.4.4. Functional Properties

Solubility and swelling power and of samples were determined according to method of [33]. Flour sample (0.6 g) was dispersed in DI water (30 mL), prior to heating in a temperature-controlled water bath at 55, 65, 75, 85, and 95 °C for 30 min with mixing interval at every 5 min. The heated samples were cooled to room temperature prior to centrifuging at 4000× *g* for 10 min. The supernatant was carefully pulled out using a syringe, dried for 24 h at 105 °C and weighed. The precipitates were immediately weighed. Solubility and swelling power were calculated according to the following equations:(3)$$\mathrm{Solubility}\;\left(\%\right)=\frac{\mathrm{Weight}\;\mathrm{of}\;\mathrm{dried}\;\mathrm{supernatant}}{\mathrm{Weight}\;\mathrm{of}\;\mathrm{dried}\;\mathrm{sample}}\times 100$$
(4)$$\mathrm{Swelling}\;\mathrm{power}\;\left(g/g\;\mathrm{dry}\;\mathrm{flour}\right)=\frac{\mathrm{Weight}\;\mathrm{of}\;\mathrm{precipitate}}{\mathrm{Weight}\;\mathrm{of}\;\mathrm{dried}\;\mathrm{sample}\times [1-\left(\frac{\mathrm{Solubility}}{100}\right)]}$$

Pasting properties of samples were investigated according to the AACC method 76–21.01 [34] using a Rapid Visco Analyzer (RVA-4, Newport Scientific, Jessup, MD, USA). Flour sample (3.0 g, dry basis) was added to the canister and filled with DI water (25 mL). The flour slurry was equilibrated for 1 min at 50 °C before being heated from 50 to 95 °C at the rate of 12 °C/min, held at 95 °C for 2.5 min, and cooled to 50 °C at the same rate. The paddle speed was set at 960 rpm for the first 10 s and then held at 160 rpm. The units of pasting properties were converted from RVU to cP by multiplying with 12 [35]. Peak viscosity, trough viscosity, breakdown viscosity, final viscosity, setback viscosity, peak time, and pasting temperature of samples were identified from the obtained viscograms using the built-in software (Thermoline for Windows^®^ v.2.2, Newport Scientific).

#### 2.4.5. In Vitro Gastrointestinal Digestion and Estimated Glycemic Index (eGI)

The rate of in vitro starch hydrolysis was analyzed according to the method recommended by [36]. Briefly, lotus seed flour (100 mg) was put into a 50 mL screw-capped test tube and 10 mL of HCl–KCl buffer (pH 1.5) was added before sample was homogenized using GENIE G560, Vortex-Genie 2 Vortex Mixer 120 V (Scientific Industries, Bohemia, New York, NY, USA) Then, 200 μL of pepsin solution containing 1 mg of pepsin from porcine gastric mucosa (≥250 U/mg solid, Sigma-Aldrich) in 10 mL HCl–KCl buffer (pH 1.5) was added into each sample. The sample was then incubated at 40 °C for 60 min in a shaking water bath. The volume was raised to 20 mL by adding 9 mL of tris–maleate buffer (pH 6.9). To start the starch hydrolysis, another 1 mL of tris–maleate buffer containing 1 mg of α-amylase type VI-B from porcine pancreas (≥10 U/mg, Sigma-Aldrich) in 25 mL was added to each sample. The sample was placed in a shaking water bath at 37 °C with moderate agitation and 100 μL of aliquots were taken every 30 min from 0 to 2 h. After completion, the tubes were placed in a boiling water bath for 5 min to inactivate the enzyme. Then, the digesta was centrifuged at 4600× *g* for 10 min to separate precipitate for RS analysis. The supernatant was further added with 0.4 M sodium–acetate buffer (pH 4.75, 1 mL) and amyloglucosidase from *Aspergillus niger* (Megazyme, 60 µL). Samples were incubated for 45 min at 60 °C. Glucose concentration in sample was then measured using glucose oxidase–peroxidase kit GOPOD reagent enzymes (Megazyme). Rate of starch digestion was expressed as a percentage of glucose at different times (0, 20, 30, 60, 90, and 120 min). The hydrolysis index (HI) was initially determined using area under the curve from GraphPad Prism 7 software (GraphPad Software, San Diego, CA, USA) and using white bread as a reference food. From the linear relationship between hydrolysis index and glycemic index, estimated glycemic index (eGI) was calculated according to the equation:eGI = (0.549 × HI) + 39.71(5)

In addition, glycemic load (GL) of the flour samples was determined from eGI and the amount of available carbohydrate in a common serving, i.e., 30 g (1/4 cup). Available carbohydrate was calculated by subtracting total dietary fibers from total carbohydrates. GL was calculated using the following equation:GL = eGI × g of available carbohydrate per serving/100(6)

### 2.5. Statistical Analyses

Statistical analyses were performed using the IBM SPSS Statistic 19 (IBM, Armonk, New York, NY, USA). All results were reported as mean and standard deviation of three replicates, except that chemical composition was reported as mean of duplicate analysis. Differences between lotus seed flour and its modified samples were determined using Analysis of Variance with Tukey’s post hoc test (*p* < 0.05).

## 3. Results and Discussion

### 3.1. Chemical Composition of Native and Modified Lotus Seed Flour

Moisture content of native lotus seed flour was 7.92% *w*/*w*, while all modified flours contained higher amount of water ranging from 10.51–12.48% *w*/*w* (Table 1). All lotus seed flour samples had moisture content complying with the standard for flour (<14% *w*/*w*). Protein content of N-LSF was 26.79% on a dry basis, which was the highest among other samples. Carbohydrate was the major composition of N-LSF, accounting for 67%, while fat and ash contents were only about 2 and 4%, respectively, on a dry basis. Partial gelatinization and pullulanase treatment caused the lotus seed flour to have slightly less protein than the native flour, while heat–moisture treatment decreased the protein content in lotus seed flour to about 12% dry basis. The preparation of HMT involved soaking of the flour, which probably led to the loss in soluble protein by leaching. Denaturation of protein during the heat moisture treatment may also have contributed to the decrease in protein content of HMT. This result is in agreement with work of Jia et al. [37] that the tertiary structure of lotus seed protein partly unfolds when subjected to heat treatment, especially heating at 100 °C for 30 min.

It is noticeable that protein content of N-LSF was significantly higher than other conventional flours, such as rice (9%), corn (8%), wheat (15%), and potato (9%) [26,38]. Moreover, our native lotus seed flour showed slightly higher protein content when compared with other Thai lotus cultivars, namely Chatchompoo (21.41%), Patoom (21.36%), Chatkaw (17.28%), and Boontaric (17.16%), as well as that of Chinese lotus seed (18.7%) [39]. Lotus seed protein has been considered a high-quality plant protein similar to soybean protein, owing to its predicted protein efficiency ratio of above 2.0 and the proportion of total essential amino acids exceeded 36% of total amino acids [40]. However, in this study the amino acid composition of lotus seed protein was not analyzed. Carbohydrate was the major component in the studied lotus seed flour, although the content was slightly lower than that of other lotus cultivars in Thailand, as reported by Singthong and Meesit [3] (63–67%).

Fat content of PG and HMT was similar to that of native flour (about 2% dry wt) but EP contained slightly lower fat than the others. This was because all modification methods did not involve any hydrophobic solvent that could lead to the change in fat content. Similar ash content of 4.3% was also observed for N-LSF and PG, while HMT and EP contained less ash at 2.33 and 1.06%, respectively. It was plausible that the decrease in ash content of HMT and EP was caused by leaching of minerals into water during those modification processes. The total carbohydrates content of HMT and EP was higher than N-LSF and PG. This finding could be explained by the lower protein and ash contents of HMT, and the lower fat and ash content of EP, when compared with N-LSF and PG.

N-LSF contained 8.36 and 3.59% of soluble and insoluble dietary fiber, respectively, on a dry basis. Slightly lower amounts of both types of dietary fibers were found in PG. Significant alteration of dietary fiber content was observed in HMT and EP. Heat–moisture treatment gave lotus seed flour with lower soluble fiber but higher insoluble fiber content. An opposite trend was observed for EP, of which most of the dietary fiber was soluble fiber and there was only <0.10 g of insoluble fiber in 100 g dry wt. Soluble components including soluble dietary fiber might have lost upon the preparation steps of HMT. In addition, it was possible that heat–moisture treatment led to the denaturation of protein in the lotus seed flour; thus, the denatured protein could form a layer on the surface of starch granules. Consequently, the protein-coated starch granules were inaccessible to the enzymes used for dietary fiber analysis and became less soluble in water. Dupuis et al. [41] demonstrated the change in protein conformation from α-helix to β-sheet upon heating at 96 °C for up to 30 min. Such conformational alteration enabled more extensive cross-linking, resulting in enhanced protein matrix strength that could diminish the susceptibility to hydrolysis. The increase in soluble dietary fiber content of EP might be explained by the fact that the debranching activity of pullulanase reduced the molecular size of insoluble dietary fiber such as β-glucan. Short linear chains obtained from pullulanase hydrolysis of glucan has been reported to improve its solubility and water-holding capacity [42]. It was observable that all flour samples, except HMT, had about similar total dietary fiber content of 11–12% dry wt.

### 3.2. Amylose Content and Starch Fractions of Native and Modified Lotus Seed Flour

Based on the colorimetric iodine method, the percentage of amylose in N-LSF was 24.30% on a dry basis (Table 2). It has been reported that starch accounted for approximately 64% dry wt of lotus seeds [2], and amylose content in lotus starch was as high as 30% [43]. Such content of amylose was comparable to that of sacred lotus seed flour in this study. Heat–moisture and pullulanase treatments resulted in the modified lotus seed flours with higher amylose content; while partial gelatinization did not cause any change. The increase in amylose content was more evident in HMT than EP. This was possibly attributed to the high thermal energy of the HMT process that cleaved the covalent glycosidic bonds and shortened the long amylopectin chains [44]. For EP, treatment with pullulanase enzyme hydrolyzed the α-1,6-glucosidic bonds and resulted in the debranching of starch. So, there were more linear fractions linked by α-1,4-glucosidic bonds that could absorb iodine into their hollow helix structure. This led to the increase in iodine adsorption, which was interpreted as amylose content. The short linear glucan chains obtained from debranching of waxy corn starch by pullulanase were also characterized as amylose elsewhere [45].

The enzymatic digestibility of native and modified lotus seed flours was investigated. RDS, SDS, and RS, which were classified consecutively by reaction times, represented the three different starch materials found in lotus seed flour samples (Table 2). N-LSF contained 6.56% RDS, 12.72% SDS, and 80.72% RS, indicating that most of the native starch was indigestible. Pullulanase treatment did not affect the RDS content, while partial gelatinization and heat–moisture treatment significantly changed the RDS of lotus seed flour. The highest RDS content of 19.53% was found in PG and the lowest amount of 4.33% was present in HMT. PG underwent gelatinization which involved the destruction of starch granules; thus, it was more susceptible to digestion by enzyme. Similar to RDS, the highest and lowest amounts of SDS were observed in PG and HMT, respectively. Starch of PG consisted of less RS than the native flour (64.46%). On the other hand, 92.52% and 86.99% of the starches presented in HMT and EP was RS. It was seemingly that RS content of the modified flour samples depended on their RDS and SDS contents. Partial gelatinization increased RDS and SDS; thus, the RS content was lower than the native flour. On the other hand, the decreases in RDS and SDS contents of HMT, as well as the decrease in SDS content of EP, resulted in their higher RS content than that of native flour. The higher amylose content of HMT and EP could also contribute to the formation of RS, particularly RS type 3 which is the retrograded amylose and starch [7]. The interactions between amylose-amylose and amylose-amylopectin chains during the heat–moisture treatment could also contribute to the increase in RS content of HMT [46]. Such rearrangement of molecular chains led to the more compact granule structure to became less susceptible to enzyme hydrolysis. For EP, the short-chain amylose chains obtained from hydrolysis of amylopectin could recrystallize to form an ordered structure, which limited the accessibility of enzyme during digestion [23]. So, modifications by HMT and EP could retard the starch digestibility of lotus seed flour. It is noteworthy that the determination method for RS was different from that of RDS and SDS, in terms of the digestive enzyme dosage, and hydrolysis condition. However, these methods have been widely used for determination of starch fractions in carbohydrate-rich food.

### 3.3. Functional Properties of Native and Modified Lotus Seed Flours

Solubility, also known as water-soluble index, of native and modified lotus seed flours were studied at 55, 65, 75, 85, and 95 °C, and the results are shown in Figure 1. At any of the temperatures, N-LSF had the highest solubility in water, followed by PG, HMT, and EP, respectively. In addition, the solubility of lotus seed flour samples seemed to increase with the temperature, especially at 85 and 95 °C. Depending on temperature, the solubility of PG, HMT, and EP was 2–3, 3–7, and 7–20 times lower than that of N-LSF, respectively. The lower solubility of lotus seed flour subjected to pre-gelatinization at 80 °C than that of native one has also been reported elsewhere [16]. This might be due to the rearrangement of double helices in the amylose region that limited the hydration and swelling of starch granules. During partial gelatinization, starch granules were partially destroyed by the penetration of water, resulting in the rearrangement of amylose that could restrict the rehydration of the gelatinized granules. For HMT, it was possible that thermal modification and destruction of long amylopectin chains occurred during the treatment, and gradually destroyed the crystalline structure of starch granules. Such changes promoted the recrystallization that impeded the solubilization [47,48,49]. For enzyme-treated flour, hydrolysis by pullulanase cut the amylopectin down to short-chain amylose that could form more crystalline regions upon recrystallization and largely diminished the solubility of starch granules [25]. The increase in solubility of flour samples at higher temperatures resulted from the more leaching of amylose from the broken starch granules. High temperatures also destroyed the hydrogen bonding within the crystalline region and destructed the crystallinity of starch granule. Consequently, there were more available hydroxyl groups to bind with water. The solubility of N-LSF and PG slightly changed across the temperatures ranging from 55–95 °C. Such finding was contradict to previous reports of Guo, Zeng, Lu, et al. [33] and Chen et al. [50], probably owing to the difference in flour composition. It is noteworthy that the difference in solubility of different flour samples in this study might also cause by their different composition (Table 1), especially water soluble components like protein and soluble dietary fiber.

The swelling power, which indicates the amount of water that can penetrate into and be adsorbed by the flour, of different lotus seed flour samples is presented in Figure 2. At 55 °C, PG showed the highest swelling power, followed by N-LSF, HMT, and EP. Partial gelatinization resulted in the lower crystallinity of the starch granules; thus, more water could penetrate and be retained in the granule. For HMT and EP, the lower swelling power than other flours was consistent with their higher amylose content (Table 2). Swelling power of starch has been attributed to a higher degree of intermolecular association and higher amylose content. The high amylose content led to the greater extent of starch retrogradation in the flour. The packed structure thus hindered the penetration of water and swelling of granules. The reduction of swelling power might also cause by the internal rearrangement of starch granules, such as interactions of amylose–amylose, amylose–amylopectin, amylose–lipid complex, and intramolecular bonding [18,26,50]. All of which resulted in the modifications of the crystalline structure that impeded the penetration of water into the starch granules. Similar to solubility, swelling power of any of the flour samples increased with the increasing temperature. At 75 °C, swelling power of EP outpaced that of HMT and was similar to that of native flour. N-LSF had the similar swelling power to PG at 85 °C, and those of HMT and EP were the same. At 95 °C, native flour exhibited the greatest swelling power, followed by PG and EP, while HMT was the least swollen.

Although swelling power is known to be harmonious with solubility [13,18], there was no direct correlation between swelling and solubility of lotus seed flour samples in this study. It is worth noting that lotus seed flour consisted of not only starch but also protein, fat, and dietary fibers (Table 1). High protein content in flour might also affect swelling power. When starch granules were embedded within the protein matrix, they were less accessible by water and thus the swelling was limited [51]. Moreover, lipid could also interact with amylose to form an amylose–lipid complex that diminished the swelling ability of the granules [2]. However, in this study, lipid content in N-LSF and the modified flour was quite low (<2% dry basis; Table 1).

Pasting properties are associated with the rigidity or swelling potential of starch granules and the leaching of amylose molecules. Figure 3 shows the representing viscograms of native and modified lotus seed flours. Pasting properties, including peak viscosity, trough viscosity, breakdown viscosity, setback viscosity, final viscosity, peak time, and pasting temperature, are compiled in Table 3. When starch was heated in the presence of water, the starch granules swelled and ruptured, resulting in gelatinization. Continuing heating of gelatinized granules led to increased viscosity due to the leaching of amylose. Peak viscosity represents the water-binding capacity in starch granules. Disruption of granules results in a decrease in paste viscosity which is termed as trough viscosity. Breakdown viscosity indicates the degree of disintegration of the granules. Setback viscosity demonstrates the tendency to retrograde. Peak time indicates the rate of granule swelling. Pasting temperature is the temperature at which the viscosity of starch begins to rise [52,53].

The results showed that all pasting parameters of N-LSF were relatively closed to those of native lotus seed starch in the literature [54]. Different modification methods gave lotus seed flours with different pasting characteristics. PG had slightly lower values of all pasting parameters than native flour, but the peak time and pasting temperature were not different from N-LSF. For HMT, the values of all pasting parameters, except pasting temperature and peak time, were substantially lower than those of native flour. Among all lotus seed flours in this study, EP exhibited the greatest values of all pasting parameters, but the lowest pasting temperature and peak time.

The pasting properties, particularly peak viscosity, peak time and pasting temperature, of native and modified lotus seed flours correlated with their swelling power (Figure 2). Flours with higher swelling power would exhibit higher peak viscosity, but lower peak time and pasting temperature. The pasting properties of lotus seed flours were also attributed to the ratio of amylose and amylopectin, as well as other constituents such as protein and lipids. The interactions between starch and non-starch component, particularly protein, could lead to the increase in peak viscosity [55,56,57,58]. Modified lotus seed flours exhibited different pasting properties because different modification processes caused different changes to the starch granules. Partial gelatinization caused the partial disruption of starch granules. So, the pasting properties of PG did not much differ from its native but the breakdown viscosity was much lower, probably because the granules had already been disrupted during the partial gelatinization process. Heat–moisture treatment could alter the crystalline structure and affect the pasting profile of flour. It has been reported that heat–moisture treatment impacted the formation of interaction among the starch components, which limited starch granule dispersion and swelling. This phenomenon reduces the availability of hydroxyl groups for hydration, causing the extensive decrease in swelling power (Figure 2) that consequently altered the pasting profile of starch (Figure 3 and Table 3) [47]. Heat–moisture treatment is known to increase the pasting temperature but decrease the peak viscosity of starch [59]. The similar pasting profile has also been observed in heat–moisture treated sweet potato flour [60]. The lower setback viscosity of PG and HMT indicated their lower retrogradation tendency than the native lotus seed flour, which might be useful for application in thickened food products like soups and sauces [61].

For EP, the short linear amylose chains, which were abundant in EP (Table 2), might leach out of the granule more rapidly, thus the gelatinization and pasting were accelerated. This was evident in the slightly lower pasting temperature, lower peak time, and much higher peak viscosity than N-LSF and all other modified flours (Table 3). EP had comparable swelling ability to HMT at 55 and 65 °C, which was lower than its pasting temperature (Figure 2). Around this temperature, the short linear amylose chains formed a highly-ordered crystalline structure that prevented the penetration of water [62]. However, at temperatures exceeding pasting temperature, the starch granules of EP fully swelled and thus the viscosity values were the highest in this case. It should be noted that EP was the only sample whose modification process did not involve heating, meaning that the starch granules were non-gelatinized. The fact that EP contained higher soluble dietary fiber than the other flour samples could plausibly be due to its high viscosity (Table 1). The increases in peak viscosity, trough viscosity, breakdown viscosity, final viscosity, and setback viscosity with the increasing soluble dietary fiber concentration have been demonstrated previously in corn resistant starch [63] and rice starch [64].

### 3.4. Estimated Glycemic Index of Native and Modified Lotus Seed Flours

In this study, starch hydrolysis was carried out in vitro to simulate the in vivo digestive characteristics of lotus seed flours and to estimate their glycemic indices. As shown in Table 4, HI of N-LSF was 11.87 with the corresponding eGI of 46.23, making it be categorized as “low glycemic food”. Partial gelatinization gave the modified flour with two-times higher HI. Consequently, the eGI of PG was higher than N-LSF and fell within the medium GI category. Such eGI value was consistent with the fact that PG contained the highest amount of RDS and the lowest amount of RS among other samples (Table 2). Other modification methods did not affect the HI and eGI of lotus seed flour, although HMT and EP contained more RS than the native flour. This was consistent with the higher amylose, RDS and total starch content of HMT and EP than the native flour. Furthermore, the ratio between amylose and amylopectin, physical entrapment of starch molecules, as well as other ingredients such as sugar, fat, proteins, dietary fiber, and anti-nutritional substances, have been reported to affect the glycemic index of food samples [65,66]. So, it was plausible there might be complex structure between starch and other components in lotus seeds, as well as the presence of natural substances that could retard the starch digestion of N-LSF. Destruction of such structure or substances might occur during the flour modification process. Therefore, the effect of flour modification on the eGI was less obvious. The GL of HMT and EP were significantly lower than that of PG and N-LSF, owing to their lower eGI and higher RS content. This indicated the lower available carbohydrate in a reference portion of flour (30 g or 1/4 cup) for HMT and EP. However, the GL of all flour samples were <10, hence they were in the same classification as low glycemic load.

## 4. Conclusions

Starch modification based on partial gelatinization, heat–moisture treatment, and pullulanase enzyme treatment significantly altered the composition and properties of lotus seed flour to different extents. Partial gelatinization did not affect the chemical composition and pasting properties of the flour but largely increased the digestibility of starch. Heat–moisture treatment resulted in the modified lotus seed flour with less protein but more amylose, insoluble fiber, and RS, which restricted its swelling and pasting behavior while maintaining the low GI value of the native flour. Treatment with pullulanase altered the properties of lotus seed flour in a similar manner to heat–moisture treatment, but the enzyme-modified flour exhibited more rapidly pasting and much larger changes in viscosity during pasting cycle than both the native and other modified flours. Based on the obtained results, heat–moisture treatment could be a potential modification method for producing low glycemic lotus seed flour that could be applied in food product without affecting the consistency of food. On the other hand, pullulanase treatment could improve the thickening effect of lotus seed flour while maintaining the low glycemic value.

## Figures and Tables

**Figure 1 foods-11-02473-f001:**
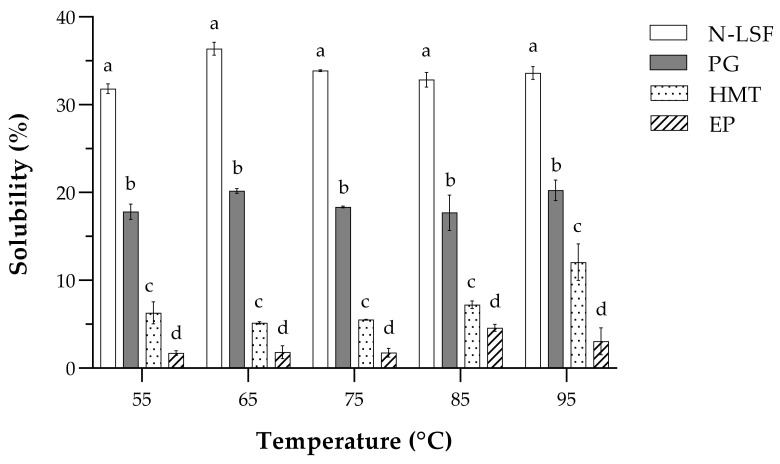
Solubility at different temperatures of native and modified lotus seed flours. Data are presented as means of three replicates with error bars of standard deviations. Open bars represent native lotus seed flour (N-LSF), filled bars represent partially gelatinized lotus seed flour (PG), dotted bars represent heat–moisture treated lotus seed flour (HMT), and striped bars represent enzymatically modified lotus seed flour (EP). Different superscripts indicated significant differences between samples at the same temperature (*p* < 0.05).

**Figure 2 foods-11-02473-f002:**
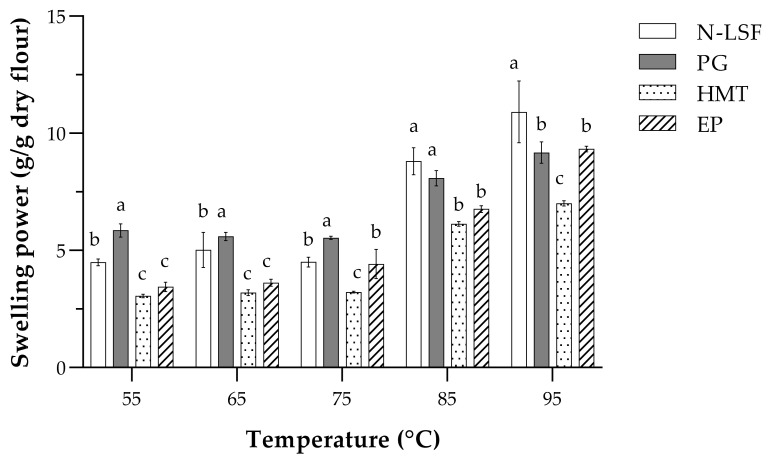
Swelling power at different temperatures of native and modified lotus seed flours. Data are presented as means of three replicates with error bars of standard deviations. Open bars represent native lotus seed flour (N-LSF), filled bars represent partially gelatinized lotus seed flour (PG), dotted bars represent heat–moisture treated lotus seed flour (HMT), and striped bars represent enzymatically modified lotus seed flour (EP). Different superscripts indicated significant differences between samples at the same temperature (*p* < 0.05).

**Figure 3 foods-11-02473-f003:**
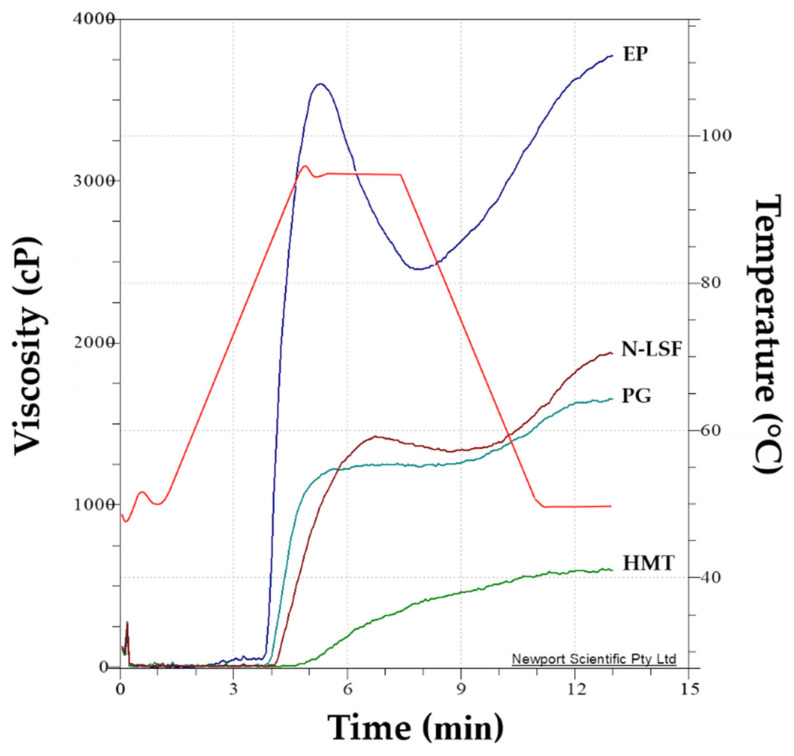
Typical viscograms of native and modified lotus seed flours. N-LSF: native lotus seed flour; PG: partially gelatinized lotus seed flour; HMT: heat–moisture treated lotus seed flour; EP: enzymatically modified lotus seed flour. Red line indicates temperature profile of the pasting cycle.

**Table 1 foods-11-02473-t001:** Chemical composition of native and modified lotus seed flours.

Sample	Moisture (g/100 g Flour)	Composition (g/100 g Dry Flour)					
		Protein (N × 5.70)	Fat	Ash	CHO	Dietary Fiber	
						Soluble	Insoluble
N-LSF	7.92	26.79	1.84	4.31	67.06	8.36	3.59
PG	12.48	25.81	2.00	4.34	67.85	7.28	3.55
HMT	11.87	11.99	1.99	2.33	83.70	5.25	9.88
EP	10.51	24.26	1.40	1.06	73.28	11.62	<0.10

CHO: total carbohydrates; N-LSF: native lotus seed flour; PG: partially gelatinized lotus seed flour; HMT: heat–moisture treated lotus seed flour; EP: enzymatically modified lotus seed flour; Data are presented as means of duplicate analysis.

**Table 2 foods-11-02473-t002:** Amylose content and starch fractions of native and modified lotus seed flours.

Sample	Amylose (g/100 Dry Flour)	Starch Fraction (g/100 g Starch)		
		RDS	SDS	RS
N-LSF	24.30 ± 0.33 ^c^	6.56 ± 0.31 ^b^	12.72 ± 1.71 ^b^	80.72 ± 1.46 ^c^
PG	26.49 ± 0.51 ^c^	19.53 ± 0.48 ^a^	16.00 ± 0.53 ^a^	64.46 ± 0.42 ^d^
HMT	39.23 ± 0.31 ^a^	4.33 ± 0.44 ^c^	3.14 ± 0.69 ^d^	92.52 ± 0.69 ^a^
EP	34.57 ± 0.48 ^b^	7.34 ± 0.20 ^b^	5.67 ± 0.37 ^c^	86.99 ± 0.49 ^b^

RDS: rapid digestible starch; SDS: slow digestible starch; RS: resistant starch; N-LSF: native lotus seed flour; PG: partially gelatinized lotus seed flour; HMT: heat–moisture treated lotus seed flour; EP: enzymatically modified lotus seed flour. Data are presented as means ± standard deviations of three replicates. Different superscripts within the same column indicated significant differences between means (*p* < 0.05).

**Table 3 foods-11-02473-t003:** Pasting properties of native and modified lotus seed flours.

Sample	Viscosity (cP)					Peak Time (min)	Pasting Temperature (°C)
	Peak	Trough	Breakdown	Setback	Final		
N-LSF	1416.67 ± 8.14 ^b^	1294.67 ± 12.06 ^b^	122.00 ± 6.24 ^b^	638.33 ± 13.01 ^b^	1933.00 ± 13.11 ^b^	6.71 ± 0.10 ^b^	86.33 ± 1.22 ^b^
PG	1237.67 ± 16.92 ^c^	1217.00 ± 10.58 ^c^	20.67 ± 6.51 ^c^	431.00 ± 3.61 ^c^	1648.00 ± 7.00 ^c^	6.80 ± 0.13 ^ab^	84.67 ± 1.75 ^bc^
HMT	308.67 ± 5.51 ^d^	290.32 ± 17.03 ^d^	5.13 ± 3.57 ^d^	397.00 ± 1.73 ^d^	592.33 ± 2.08 ^d^	7.00 ± 0.00 ^a^	94.92 ± 0.14 ^a^
EP	3473.68 ± 49.52 ^a^	2372.96 ± 40.29 ^a^	1100.64 ± 29.76 ^a^	1231.32 ± 15.47 ^a^	3604.36 ± 30.59 ^a^	5.22 ± 0.08 ^c^	82.70 ± 0.98 ^c^

N-LSF: native lotus seed flour; PG: partially gelatinized lotus seed flour; HMT: heat–moisture treated lotus seed flour; EP: enzymatically modified lotus seed flour. Data are presented as means ± standard deviations of three replicates. Different superscripts within the same column indicated significant differences between means (*p* < 0.05).

**Table 4 foods-11-02473-t004:** Estimated glycemic index and glycemic load of native and modified lotus seed flours.

Sample	AUC	HI	eGI	GI Category	GL	GL Category
N-LSF	269.30 ± 23.62 ^b^	11.87 ± 1.04 ^b^	46.23 ± 0.57 ^b^	Low	8.50 ± 0.11 ^c^	Low
PG	841.13 ± 64.47 ^a^	37.08 ± 1.50 ^a^	60.07 ± 0.82 ^a^	Medium	12.91 ± 0.90 ^a^	Low
HMT	596.87 ± 23.05 ^ab^	17.63 ± 0.68 ^b^	49.39 ± 0.37 ^b^	Low	12.14 ± 0.21 ^a^	Low
EP	541.83 ± 10.33 ^ab^	16.00 ± 0.30 ^b^	48.50 ± 0.17 ^b^	Low	10.55 ± 0.05 ^b^	Low

AUC: Are under the curve; HI: hydrolysis index; eGI: estimated glycemic index; GL; glycemic load; N-LSF: native lotus seed flour; PG: partially gelatinized lotus seed flour; HMT: heat–moisture treated lotus seed flour; EP: enzymatically modified lotus seed flour. Data are presented as means ± standard deviations of three replicates. Different superscripts within the same column indicated significant differences between means (*p* < 0.05).

## Data Availability

The authors confirm that the data supporting the findings of this study are available within the article.

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
