# Peer review of "Effect of Physical and Enzymatic Modifications on Composition, Properties and In Vitro Starch Digestibility of Sacred Lotus (Nelumbo nucifera) Seed Flour"

_foods, 2022, doi:10.3390/foods11162473_

Round 1

Reviewer 1 Report

The study from manuscript foods-1754550 has been well designed, and the results are well described and discussed. Some minor comments to improve your document are included below.

L28 Keywords – Please replace those from the title

L31-34 Introduction – More recent data on Diabetes is accessible from International Diabetes Federation (https://idf.org/aboutdiabetes/what-is-diabetes/facts-figures.html), Diabetes atlas (https://diabetesatlas.org/).

L65-89 This paragraph is too big. The authors must split it and include the corresponding references from the statements provided from L65 to L71. It is known that References 17 and 18 are reviews on the modification of starch, and the corresponding articles regarding each modification method must be indicated. It is not reasonable that all that information comes from references 17 and 18, as they present a compiled of articles.

L78-79 – The authors cannot provide statements from their results in this section. If the information in these lines comes from any reference, the authors must adequately cite them. Otherwise, if that comes from your results, it must be indicated as a suggestion or possibility. The introduction includes a background, not results.

L137-142 – Please indicate the corresponding numbers of each AOAC method used.

L256 – Correct the citation to “Singthong and Meesit [3] (63-67%)”.

L257 – With “unpresented data” do the authors mean “unpublished data”?

L169 and L183 – Correct “4000xg” to “4600×g”

Figure 1 – Please correct the degree symbol on the X-axis.

Conclusion – Remove “In this study.” I suggest using “Starch modification methods based on (…) significantly modifies the composition and properties of lotus seed flour”.

Reviewer 2 Report

- The manuscript is well written. And the experiments are well planned and explained.

-Introduction gives sufficient background on why the work was taken. 

-Discussion is well written, and references are cited as per the discussion.

- The figures and tables are clear. 

- GI values of different foods(listed) if available should be mentioned in line no. 46 and 47

- Why >50% reduction in protein in HMT, any specific reason. This can be mentioned in the discussion. 

The manuscript is suitable for publication after minor corrections.

Reviewer 3 Report

In the manuscript submitted, the authors studied effects of physical and enzymatic modifications on composition, properties and in vitro digestibility of sacred lotus. The  topic selection has certain novelty. Meanwhile, authors also gave in-depth analysis and discussion. However, the manuscript contain less novel findings and the results are easy to be predicted. Specific comments are as follows.

1. Firstly, the object of the study is flour rather than starch, so in the interpretation of a large number of experimental results, authors only analyzes the structural changes of starch itself, but lacks the analysis on the influence of other components in flour.

2. Line 19 Please rewrite the sentences chemical composition of HMT and EP but not of PG differed from the native flour, particularly protein and dietary fibers”

3. Line107: boiled at 75 ℃?Partial gelatinization(Line 105) and achieve gelatinization

4. Line118: For Heat-moisture treatment, what is the basis for choosing 100

5. Line 191: The usual method is to add deionized water to reach a certain total mass. Therefore, how to ensure consistent sample concentration?

6. The data in Figure 3 do not match the data in Table 3. For HMT sample curve, why is there peak viscosity? No peaks can be observed.

Reviewer 4 Report

In this study, composition properties and in vitro digestibility of lotus seed flours (LSF) after physical and enzymatic modifications were compared.  Using LSF as material, the author investigated the digestibility changes of starch in LSF after physical and enzymatic treatment. From the composition of LSF after different treatments, the protein, soluble fiber and insoluble fiber contents of LSF after different treatments changed significantly. However, the total starch content and lipid content of LSF, which are directly relevant to the discussion in this paper, are not provided.  Moreover, the correctness of some data needs to be further confirmed.  For the above reasons, a major revision is required of this manuscript.

1.   Line 107: boiled? 75ºC?  Which temperature is used for this treatment?

2. Section 2.2.2 & 2.3:  What is the recovery of LSF after these two treatments?

3.  Line 141-142: What is the literature for this method?

4. Line 144: “iodine adsorption method” should be “iodine absorption method”.

5.  Table 1

A. According to Section 2.6, the data should provide standard deviation.

B. Total starch and lipid content of LSF should be provided

C. Line 272-274: What is the composition of insoluble dietary fiber? How does pullulanase cause the degradation of insoluble dietary fiber?

6. Line 284-286: Increase in “iodine absorption” or “amylose”? What is definition of “amylose”?  Can the branched chains obtained from the pullulanase degradation of amylopectin be called “amylose”?

7. Table 2: What does the dry weight basis in the table mean? Dry flour weight?  From Table 1, the content of protein and dietary fiber in different samples are significantly different. And, the digestive investigation of LFS in this study is focus on starch, so the data in Table 2 should be based on starch. Therefore, it is very important to provide data on the total starch content of LFS.

8. Line 328: How is amylose leached from LFS at the moisture content of HMT?

9. Line 370: Lipid content is not provided in this manuscript.

10. Figure 3 & Table 3

A. Figure 3: Native should be N-LSF. Abbreviations should be consistent.

B. As can be seen from Figure 3, EP has a similar pasting temperature to that of native and PG, but the results in Table 3 show that the pasting temperature of EP is more than 15°C higher than that of native and PG. The results of the graph and the table obviously do not match, and the correctness of the data should be reconfirmed, and relevant discussion should be revised.

11. Line 309-312:  However, the data in Table 1 show that the protein content of the HMT samples is the lowest.  Thus, this hypothesis requires more obvious evidence.

12. Table 4: Except for increased eGI of PG, the eGI of modified LSF was similar to that of untreated ones.  The authors should elaborate on the contribution of the different modifications to digestibility of LSF.

Round 2

Reviewer 4 Report

1.  Table 1

A.  The data in this form should be reconfirmed.  Why does the total of chemical compositions (including protein, dietary, fat, ash and CHO) significantly exceed 100% (106%)?

B.  The chemical compositions of the treated samples (such as PG, HMT and EP) should also be fully provided.

2. Table 2

A.  The author proposes completely different starch digestion data from the previous version and cannot judge the correctness of the data.

B.  This table is to discuss the digestibility of starch, so the relevant data should be based on starch.

C. RS and SDS/RDS in the table were determined by two different methods.  The enzyme dose used for these two methods is different and reaction conditions are also different.  Therefore, the discussion of RDS, SDS and RS should consider the problems caused by different methods.  In addition, it is unreasonable to use the total content of RDS, SDS and RS as the total starch content.

D. In general, the CHO content of a food product is considered to be mainly composed of starch. The CHO of N-LFS in Table 1 is 61%, while the total starch content of N-LFS in the table is 18.7%. Why is there a difference of more than 40% between the two determinations? 

E. It is necessary to further confirm the correctness and reasonableness of the data presented in this table.

Author Response

Table 1

The data in this form should be reconfirmed.  Why does the total of chemical compositions (including protein, dietary, fat, ash and CHO) significantly exceed 100% (106%)?

Response: We have reconfirmed the correctness of data. The carbohydrate content was calculated by subtracting the contents of moisture, protein, fat and ash from 100, according to the AOAC official method. CHO content in the proximate analysis thus also includes dietary fibers. Dietary fiber contents were analyzed separately according to the AOAC official method. For each sample, the analyses of proximate composition and dietary fibers were performed in duplicates and the means of duplicate are presented. So, the sum of moisture, protein, fat, ash and CHO in the table may not be exactly 100. We have rearranged the table for better clarity.    

The chemical compositions of the treated samples (such as PG, HMT and EP) should also be fully provided.

Response: The proximate composition of PG, HMT and EP was analyzed as suggested.

Table 2

The author proposes completely different starch digestion data from the previous version and cannot judge the correctness of the data.

Response: We decided to report the starch fraction data based on 100 g starch, as reported in the original manuscript. The correctness of amylose content and starch fractions has been checked.

This table is to discuss the digestibility of starch, so the relevant data should be based on starch.

Response: We decided to report the starch fraction data based on 100 g starch, as reported in the original manuscript.

RS and SDS/RDS in the table were determined by two different methods.  The enzyme dose used for these two methods is different and reaction conditions are also different.  Therefore, the discussion of RDS, SDS and RS should consider the problems caused by different methods.  In addition, it is unreasonable to use the total content of RDS, SDS and RS as the total starch content.

Response: The calculation for total starch content, added in the 1st revision, was removed. Statements discussing the difference in the methods used for determination of RDS/SDS and RS were added. However, the determinations were conducted according to the standard methods that are widely used and accepted.

In general, the CHO content of a food product is considered to be mainly composed of starch. The CHO of N-LFS in Table 1 is 61%, while the total starch content of N-LFS in the table is 18.7%. Why is there a difference of more than 40% between the two determinations? 

Response: The calculation for total starch content, added in the 1st revision, was removed. The CHO content of the modified flour samples was determined from their proximate composition.

It is necessary to further confirm the correctness and reasonableness of the data presented in this table.

Response: We have rechecked for the correctness and modified the presentation of data.
